# Physico-Mechanical Evaluation of Geopolymer Concrete Activated by Sodium Hydroxide and Silica Fume-Synthesised Sodium Silicate Solution

**DOI:** 10.3390/ma16062400

**Published:** 2023-03-17

**Authors:** Blessing O. Adeleke, John M. Kinuthia, Jonathan Oti, Mansour Ebailila

**Affiliations:** 1Faculty of Computing, Engineering and Science, University of South Wales, Pontypridd CF37 1DL, UKjonathan.oti@southwales.ac.uk (J.O.); 2Department of Civil Engineering, Faculty of Engineering, Bani Waleed University, Bani Waleed 238, Libya

**Keywords:** alkali-alkaline activator, GGBS, silica fume, consistency, compressive strength, tensile splitting strength, aluminosilicate, geopolymer

## Abstract

Commercial sodium hydroxide (NaOH) and sodium silicate (SS) have remained two of the leading alkaline activators widely used in producing geopolymer concrete, despite some identified negatives regarding their availability and additional CO_2_ emissions relating to the overall manufacturing process. This study reports the viability of developing geopolymer concrete using a laboratory-synthesised silica fume (SF)-derived SS solution in combination with NaOH at a molarity of 10M as an alternative binary alkali-alkaline activator to Ground Granulated Blast Furnace slag (GGBS). The use of SF in the development of geoolymer activators will pave the way for the quality usage of other high-silica content by-products from nature, industry, and agriculture. In the currently reported proof of concept, four geopolymer concrete batches were produced using different alkaline activator/precursor-A/P ratios (0.5 and 0.9) and SS to NaOH-SS/SH volume ratios (0.8/1.2 and 1.2/0.8), to establish the impact on the engineering performance. Two controls were adopted for ordinary and geopolymer concrete mixes. The engineering performance was assessed using slump and compaction index (CI) tests, while the Unconfined Compressive Strength (UCS) and tensile splitting (TS) tests were measured at different curing ages in accordance with their appropriate standards. The results indicated a reduction in slump values as the A/P ratio decreased, while the CI values showed a reversal of the identified trend in slump. Consequently, mix GC2 attained the highest UCS strength gain (62.6 MPa), displaying the superiority of the alkali activation and polymerisation process over the CSH gel. Furthermore, the impact of A/P variation on the UCS was more pronounced than SS/SH due to its vital contribution to the overall geopolymerisation process.

## 1. Introduction

Concrete is a highly “versatile construction material” and is the most widely used construction material globally. It is the most used artificial material in existence and is composed of Portland cement, aggregate, sand, and water in a specified mix proportion [1], with cement being one of the most vital ingredients. Researchers estimated that about 4800 metric tons of cement would have been produced across the world by 2030, resulting in enormous energy consumption and carbon dioxide (CO_2_) gas emissions into the atmosphere during its manufacturing stage, while 2.8 tons of raw materials would have been utilised for every ton of cement production [2,3]. As such, an enormous effort has been made by researchers to explore the possibility of completely phasing out cement by developing an eco-friendly/alternative geopolymer binder for application in concrete [4,5,6,7].

Geopolymers are inorganic alumino-silicate polymers with a three-dimensional network, which are obtained by the reaction of rich aluminosilicate materials, such as metakaolin, fly ash, Ground Granulated Blast Furnace slag (GGBS), etc., with a highly alkaline aqueous solution [7,8]. This reaction is an alkali activation mechanism where the final products of geopolymerisation are usually influenced by the chemical composition of the alkaline activators and the source aluminosilicate materials that act as a precursor [6]. Compared with traditional ordinary Portland cement (OPC), geopolymers could offer comparable performance and possess additional benefits of reduced CO_2_ emissions and energy consumption [9]. Based on industrial relevance, production, and the properties of Alkali-Activated Materials (AAM), several researchers have frequently reported the use of commercially available sodium hydroxide (NaOH) and sodium silicate (SS) as alkaline activators [10,11,12,13]. However, the cost and availability of commercially available NaOH could be an issue in some parts of the world where geopolymer concrete is required since NaOH is produced using a fixed chemical composition. According to Sun et al. [10], SS is a generic term involving a series of alkaline silicate substances with the chemical formula—Na_2x_SiO_2+X_, such as sodium metasilicate—Na_2_SiO_3_, sodium orthosilicate—Na_4_SiO_4_, and sodium pyrosilicate—Na_6_Si_2_O_7_. These compounds are generally colourless, transparent solids or white powders that are soluble in water in various amounts to form alkaline solutions [14].

SS is the most important type of silicate due to its availability and low-cost silica source, resulting in its wide use in various industrial applications such as deflocculates, emulsifiers, adhesives, detergents, and sealants [15]. Firstly, it is essential to note that the properties of commercial SS solutions are usually typified by the SiO_2_ content and molarity (M)—defined as the molar ratio between SiO_2_ and Na_2_O [10]. Secondly, the alkaline activator/precursor (A/P) ratio and the sodium silicate to sodium hydroxide (SS/SH) ratio are key parameters that impact the engineering efficiency of geopolymer concrete [16]. Shilar et al. [17] reviewed several research works on geopolymer concrete data for molarity variations ranging from 5.8 to 18M, with A/P ratios ranging from 0.30 to 0.5 SS/SH ratios indicate that increasing the molarity and A/P ratios results in the strength development of geopolymer concrete up to a specific limit.

Silica fume (SF) is a by-product from the ferrosilicon industry with tiny particle sizes and is mostly used as filler. Consequently, it comprises a high percentage of amorphous silicon dioxide, rendering it a highly reactive pozzolanic material [18]. Using SF does not only increase concrete strength, but its fine particle size also improves the cohesiveness between the binder and aggregate. This results in a reduction in bleeding rate and potential material segregation in the fresh concrete state [19]. Recent studies show that silica fume can be used as a source of reactive silica to synthesise alternative water glass (AWG)-based alkali activators in building applications [13,20]. The production of silica fume-NaOH activator is based on the high solubility of amorphous silica in NaOH solution at low temperatures, which is helpful in low-climate regions of the world [20].

Although the choice of using commercially produced SS has been well received, there has not been a perfect evaluation of the effectiveness of a laboratory-synthesised SS that can be developed and used to develop geopolymer concrete on a construction site at normal atmospheric conditions (20 ± 2 °C). This hinders the practical application and promotion of geopolymer concrete since the production of NaOH and SS is commercially controlled under very strict industrial and atmospheric conditions that are not applicable on a construction site. Raza and Zhong [5] also reiterated some additional material and transportation impacts since most geopolymer activators and precursors are generally obtained from industrial by-products with no well-established single source. Therefore, since the transportation of raw materials produces 7–8% of the total emissions of their manufacturing process [21], obtaining geopolymer materials from various sources and taking them to be produced in a single controlled factory elevates their negative transportation impacts. Thus, this paper evaluates the viability of developing a geopolymer concrete using a laboratory-synthesised SF-derived SS solution in combination with NaOH as an alternative binary alkali-alkaline activator to GGBS at normal atmospheric conditions (20 ± 2 °C) and examines the impacts of varying A/P and SS/SH ratios on the engineering performance of the developed geopolymer concrete. Additionally, this research aims to add to the advantages of geopolymer concrete with respect to low carbon emissions, sustainable improvements in construction, recycling industrial wastes, and reducing global warming potential [22].

Once the concept of a viable, strong, and durable geopolymer formulation is proven using SF in a laboratory-synthesised activator, it is hoped that this will be a step in widening scope by enthusing the use of other high-silica content waste and by-product materials from nature, industry, and/or agriculture in a similar manner. For example, high-silica content shales and slates and industrial and agricultural waste materials such as ashes from furnaces and the growing of coffee, palm oil, sugarcane, and rice, among others, would be used in a more valorised form. Their current bulk usage in cementitious systems would gradually change, as these ashes would now be used in more valuable, smaller dosages as activators rather than as target bulk materials in cementation. In addition, there is also the diversification of the routes towards geopolymer cementation, which is still in the early stages of exploration for wider industrial use. Therefore, the proof of concept reported in the current work is seen from these perspectives.

## 2. Materials

The potential performance of a developed system is best understood through detailed characterisation of the material components. The current study utilised ordinary Portland Cement (CEM-II/ B-V 32.5R), Ground Granulated Blast-furnace Slag (GGBS), silica fume (SF), sodium hydroxide (NaOH), and aggregates. OPC was supplied and manufactured by Lafarge Cement UK in accordance with BS EN 197-1:2011 [23]. Table 1 shows the oxide compositions for OPC, GGBS, and SF, while Table 2 and Figure 1 present their physical properties and Particle Size Distribution (PSD), respectively. The TG/DTG analysis of as-received cement (Figure 2) indicated a negligible mass reduction (2.8%), as the temperature increased to 1000 °C. This mass loss is represented in the DTG curve by three endothermic peaks in the temperature range of 50–150 °C, 400–450 °C, and 650–750 °C. The first peak is attributed to the dehydration of gypsum, while the second and third peaks are attributed to the dehydroxylation of portlandite (Ca(OH)_2_) and the decomposition of calcite (CaCO_3_).

GGBS was used as an aluminosilicate precursor material in this study and was manufactured in accordance with BS EN 15167-1:2006 [24] by Civil and Marine Slag Cement Ltd., Llanwern, Newport, UK. The TG analysis (Figure 2) revealed that, as the temperature increased to 1000 °C, the mass of GGBS was reduced by 2.26%. This reduction was represented in the DTG by one major peak in the temperature range of 550–700 °C, due to the decomposition of calcite. The SF used was a commercial reactive microsilica in the form of a light grey amorphous powder with a silicon dioxide content (SiO_2_) of 97.1%. It was manufactured in Norway by Elkem Silicon Materials and supplied under the trading name of Elkem Undensified Microsilica 971 by Tarmac Cement and Lime Company, Buxton Lime and Powders, Derbyshire, Derby, UK. The TG/DTG analysis revealed that the silica fume showed a negligible weight loss as the temperature increased to 1000 °C, accounting for a maximum mass loss of 1.9%. This mass loss is represented in the DTG curve by a straight line with a very light peak in the range of 400–800 °C, which could be due to the decomposition of the impurity in this product. The commercial NaOH used was in the form of white pellets with a molecular weight of 40 g/mol and a pH value of 14, and was obtained from Fisher Scientific Ltd., Loughborough, Leicestershire, UK. It was employed to produce a NaOH solution with a molarity of 10M. This obtained solution was then used as an activator to dissolve silica fume and produce a SS solution with a SiO_2_/Na_2_O molar ratio of about 2.

The coarse aggregate used was of two grades (20 mm and 10 mm) and one type of fine aggregate. The coarse aggregates were limestone aggregates, whereas the fine aggregate was natural river sand dredged from the Bristol Channel. Both coarse and fine aggregates were supplied by a local quarry through a local supplier, in accordance with the requirements of BS EN 12620:2002+A1:2008 [25]. Figure 1 shows the particle size distribution, whereas Table 3 presents some physical properties obtained in accordance with the relevant standard.

## 3. Methodology

### 3.1. Mix Design

Two different approaches were investigated to design the concrete formulations assessed in this study (Table 4). Firstly, this was to investigate the efficacy of using silica fume-derived sodium silicate solution in combination with sodium hydroxide as an alternative binary alkali-alkaline activator for geopolymer concrete production. Secondly, it was to examine the effect of two variables (the alkaline activator/precursor (A/P) ratio and the sodium silicate to sodium hydroxide (SS/SH) ratio) on the properties of geopolymer concrete. Furthermore, the mix codes for all the concrete formulations are compositionally fixed at a binder:sand:aggregate ratio of 1:2:3, where the mixes differ only in terms of the type of binder and water content.

For completeness and proper analysis, the study adopted two controls (C and GC0) for both ordinary and geopolymer concrete mixes. The ordinary concrete mix (C) was formulated using OPC at a water/binder ratio of 0.55, while GC0 was designed by activating GGBS with a binary activator blend of SS and SH solution at an AP mass ratio of 0.7 and a SS/SH volume ratio of 1/1. As for the other four geopolymer batches (GC1–GC4), they were developed using different A/P ratios (0.5 and 0.9) and SS/SH volume ratios (0.8/1.2 and 1.2/0.8). This was to establish the independent effect of these two variables (A/P and SS/SH volume ratios) on the performance of geopolymer concrete. Likewise, in all these geopolymer concrete batches, the total binder content (precursor + activator) was set equal to the mass of cement in the control plain mix, while the water content added was set to be 1.5 L to achieve a minimum slump value of 50 mm for the case of GC0.

### 3.2. Alkali-Alkaline Activator and Test Sample Preparation

The alkali-alkaline activator used in geopolymer formulations was a mixture of SH and silica fume-derived SS solution in three different SH/SS volume proportions (0.8/1.2, 1/1 and 1.2/0.8). This binary blended activator was used due to the lower cost and availability of sodium-based activators compared to potassium-based activators [17]. The SH solution was prepared by dissolving 400 g of sodium hydroxide pellets per 1000 mL of deionised water to obtain a solution with the desired molarity of 10M. In contrast, the SS solution was designed using Equation (1) and prepared by dissolving 618 g per 1000 mL of SH solution with a molarity of 10M. Furthermore, a SiO_2_/Na_2_O molar ratio of 2 and the purity of SF used in the current study (97.1 % SiO_2_) were considered.
2SiO_2_ + 2NaOH → Na_2_O(SiO_2_)_2_ + H_2_O (1)

The SH and SS solution was prepared and stored in a sealed container at an ambient temperature of 20 ± 2 °C for 24 h prior to its subsequent use as an activator for the aluminosilicate precursor. This was carried out to achieve chemical equilibrium of the exothermic reaction of SH and to complete the dissolution of silica fume particles. This is an important step, as the exothermicity of the activator can accelerate the geopolymerisation process, thus adversely impacting the properties of geopolymer concrete in its fresh state [16,26]. The densities for the SH and SS solutions were established at 1.33 g/mL^3^ and 1.59 g/mL^3^, respectively, after 24 h. In addition, the viscosity of both solutions was determined as 25 ± 2 cSt for SH and 745 ± 5 cSt for SS, using a Fungi Lab Digital Rotational Viscometer Premium Series at a speed of 100 RPM for a total period of 1.5 min.

### 3.3. Preparation of Geopolymer Concrete Specimens and Testing Methods

A mixing regime was initially employed to prepare the geopolymer concrete. Firstly, the mass of dry ingredients (cement, precursor, fine and coarse aggregates) was precisely weighed in accordance with the proportions in Table 4 and mixed in a mechanical mixer for 2 min to obtain a homogenous dry reactant mixture. Subsequently, the pre-sealed amount of SS and SH solution was gradually added to the dry mixture and mixed for two minutes, followed by the inclusion of additional water and final mixing for two minutes to produce the wet concrete.

The consistency of the fresh concrete was measured using the slump (S) and Compaction Index test (CI) according to BS EN 12350-2:2019 [27] and BS EN 12350-4:2019 [28], respectively. Thereafter, three cubes (100 mm × 100 mm × 100 mm) and two cylindrical (100 mm diameter × 200 mm height) test specimens were produced for each batch mix in accordance with BS EN 206:2013+A2:2021 [29]. The specimens were sealed with a plastic film to avoid evaporation, stored at an ambient temperature of 20 ± 2 °C for 24 h, and allowed to cure in a sealed container at a temperature of 20 ± 2 °C and a humidity of 90% [30]. After that, the test specimens were demolded after 24 h. This curing strategy was adopted instead of the heat curing condition (the optimum curing condition for geopolymers) due to its impracticability in cast-in situ construction. In addition, the adopted curing strategy is encouraged as it would eliminate any exterior heat application, thereby reducing the cost and simplifying the practical implementation of geopolymer products [31]. The hardened characteristics of all the concrete specimens were assessed in terms of Unconfined Compressive Strength (UCS) and tensile splitting strength (TS) in accordance with BS EN 12390-3:2019 [32] and BS EN 12390-6:2009 [33] at 3, 7, 14, and 28 days curing. The reported results are the average of the three cubes and two cylinder specimens for each batch mix composition.

## 4. Results and Discussions

### 4.1. Consistency of Fresh Concrete

Figure 3 presents the consistency results obtained from the slump and degree of compactability tests. The control (OPC)–C concrete attained a higher slump value of 75 mm, relative to that of 50 mm for the control geopolymer concrete (GC0). However, the trend was in reverse order in the case of compactability degree, confirming the accuracy of the laboratory investigation. The difference between these two mixes (C and GC0) could be due to the quantity/viscosity of the liquid used in these mixes and the kinetic reaction involved in each system.

In the control OPC system, the binder/water ratio was set at 0.55, implying 4 L of water was needed for 7.3 kg of cement (Table 4). As previously understood, the main reaction mechanism in forming ordinary/plain concrete is the hydration reaction between cement and water. This reaction is exothermic; thus, a higher moisture content was needed to produce workable concrete. Likewise, in geopolymer formulations, the liquid (alkaline activator) used has a higher viscosity than the water used in OPC concrete [34] and produces a stickier and more cohesive system [35]. Furthermore, material fineness could be attributed to the variation in consistency [36]. In the current study, the binder was composed of both precursor (GGBS) and activator (SS and SH), which implies that the quantity (4.3 kg) of dry cementitious materials (fineness) was lower in the case of geopolymer concrete compared to the OPC mix. In addition, GGBS is known to reduce water demand in traditional/normal concrete mixes due to its smoothness and lower water absorption [36]. Therefore, all these could be suggested to have contributed to the higher moisture content (4 L) demand for OPC compared to the 1.5 L water demand for the geopolymer concrete formulations.

The impact of A/P ratios observed in GC0, GC1, and GC2 mixes showed that the slump value decreased as the A/P ratio decreased, while the compatibility degree indicated a reversal of the identified trend in slump. This implied that reducing the activator (liquid) and increasing GGBS content induced a reduced consistency of geopolymer, which is analogous to the effect of the water/cement ratio on the consistency of OPC concrete [15] and in line with the findings of Heah et al. [37] and Yaseri et al. [38]. In addition, this decreasing trend associated with the slump values could be related to the reduction in the activator (liquid) content [4]. Therefore, the decrease in activator content initiates higher friction between the particles of the dry material, resulting in a sticky system [38], thus reducing the overall consistency [39]. Furthermore, the increase in the quantity of dry materials induces a high water retention property in the developing system [40], resulting in a higher liquid demand to lubricate these dry material particles. Additionally, it is essential to note that the excessive amount of calcium ions released from GGBS and its subsequent interaction with the alkaline activator to precipitate as calcium silicate hydrate (C-S-H) is a possible contributing factor to the reduced consistency [8]. This is because such an action accelerates the hydration ratio and speeds up the setting time of the mixture by creating further heating [41]. Consequently, the heat would trigger water consumption, which subsequently increased the viscosity of the entire system [42] and minimised the effect of particle dispersion that ultimately reduced the consistency [39,41].

Regarding the increase in the sodium silicate to sodium hydroxide (SS/SH) ratio, observation showed that the slump decreased as the SS increased, indicating the negative effect of SS on the consistency of concrete. According to Gomaa et al. [43] and Provis and Bernal [44], the identified reduction in consistency could be attributed to the higher viscosity of SS (745 ± 5 cSt) compared with 20 ± 2 cSt for SH, of which the former tends to make a cohesive, stiff, and thick mixture [6], thereby reducing the consistency [45]. This argument supports the choice of potassium silicate as an activator constituent in terms of consistency, as it possesses a viscosity value ten times lower than sodium silicate [17]. The increased SS/SH ratio also induces a higher soluble silica content, which subsequently accelerates the polymerisation degree and thus raises the overall viscosity of the system [41]. As reported by Nath and Sarker [6] and Siyal et al. [46], the higher SS dosage affects the viscosity by accelerating the dissolution of the precursor materials, which subsequently alters the condensation process and accelerates the setting time further. Overall, it can be suggested from the consistency results and BS EN 206:2013+A2:2021 [29] that all the concrete formulations can be classified as S2 standard mixes, except GC2, which is classified as an S1 dry mix.

### 4.2. Concrete Strength Development

#### 4.2.1. Unconfined Compressive Strength (UCS)

Figure 4 presents the variation in Unconfined Compressive Strength (UCS) development for all the concrete mixes over an ambient curing period of 28 days. Generally, a steady increase in strength development was observed for all the formulated geopolymer and ordinary concrete mixes over the 28 day curing period. Additionally, all the formulated geopolymer concrete mixes (GC0–GC4) achieved higher UCS at all curing age compared to the OPC (Control 1) mix. For example, a 14 day UCS for the geopolymer concrete mixes indicated a range of 35–63 MPa compared to the OPC (Control 1) mix, which yielded a 27 MPa UCS.

The display of strength gain by the OPC (Control 1) mix was due to the initial hydration of the cement components with water to produce cementing gels (C-S-H) necessary for strength gain. In contrast, the trend of strength increase in the geopolymer concrete was due to a polymerisation process that involves the chemical reaction of aluminosilicate minerals (GGBS) under alkaline conditions (SS/SH solution), resulting in a three-dimensional amorphous aluminosilicate matrix that exhibits strength corresponding to or superior to the OPC concrete [6,8]. Moreover, the variation in strength gains for each geopolymer concrete mix observed at each curing age could be attributed to a varied mix composition that impacts the alkali activation and polymerisation processes of the developed concrete system. Nath and Sarker [6] agree with the earlier hypothesis as they suggest that the alkali activation mechanism is still unclear; thus, the final products of geopolymerisation are commonly influenced mainly by the chemical composition of the geopolymer mixes (alkali activators and source materials).

The effect of the A/P ratio observed in GC0, GC1, and GC2 mixes showed that the UCS value of the geopolymer concretes increased gradually as the A/P ratio decreased. This implies that the A/P has a similar role to that observed in the impacts of water/cement ratio variation on the UCS performance of OPC concrete [47]. Hence, at the end of the 28 day curing period, there was a corresponding increase in UCS of 38.5, 47.6, and 62.6 MPa for the developed geopolymer concrete mixes—GC1, GC0, and GC2, respectively, as the A/P ratio decreased from 0.9, 0.7, and 0.5. This trend is in line with the findings of Abdulrahman et al. [4] that variations in the A/P ratio mainly govern the development of the compressive strength of geopolymer concrete. Therefore, the underlying mechanism for this action (variations in the A/P ratio) is that adding more GGBS increases the calcium ion concentration and glass content of the precursor material [48]. Thus, enhancing the ratio and extent of dissolution of calcium ions [41]. In addition, it is pertinent to note the vital role of calcium ions as a network modifier, as it leads to more rapid dissolution of the precursor [42] and permits the formation of further cementitious hydrates (CSH) or amorphously structured polymeric products via polymerisation [41]. These produced hydrates induce heating during the reaction process [49], which subsequently promotes water dehydration and accelerates the setting time [38]. Chindaprasirt et al. [49] attributed the setting time and heat generation as key parameters that influence the acceleration of the geopolymerisation process. Furthermore, the accelerated geopolymerisation reaction forms additional geopolymer gels to fill the pores within the concrete, subsequently inducing a pore-blocking effect [2]. The blockage of pores enhances a reduction in porosity, system densification, and interlocking enhancement, all of which contribute to a higher degree of strength [50].

In contrast to the reduction in UCS trend observed with an increase in the A/P ratio, observation showed an increase in consistency achieved by GC1 (Figure 3a,b) for every rise in the A/P ratio. Xie et al. [41] credited this phenomenon as a result of the excessive activator content that increases the water content within the system, thus influencing the H_2_O/Na_2_O and reducing the alkalinity (pH) value of the system. Therefore, the polymerisation process is hindered by promoting the increase of poorly polymerised reaction products. After that, the geopolymerisation process is impacted by initiating a reduced setting time and subsequent polycondensation (hardening) process [6]. However, it is worth noting that this negative water effect is different from the OPC-based system’s mechanism, in which part of the initial water employed for mixing the OPC system is chemically bonded in the hydrates [41]. During the hardening process (polycondensation phase) of the developed geopolymer concretes (alkaline-activated system), the water involved in the polymerisation phase is returned to the system. This released water remains trapped between the hydrated products formed during geopolymerisation [47]. The trapped water halts the chemical reaction ratio, causing prolonged setting times of the developed system [38], lowering the subsequent polycondensation (hardening) process [6], and ultimately producing reduced compressive strength. In addition to the negative effect of excessive water on the hardening process (polycondensation phase), excess alkaline activator (due to an increase in the A/P ratio) could remain within the pores of the geopolymer matrix owing to the difficulty of complete consumption during the geopolymer reaction.

Similar to OPC-based concrete, geopolymer concrete is a porous product; hence, alkaline compounds can be mobile through the pore network. Thus, the migration of liquid could act as an avenue for ion leaching and become a source of ion transportation from the pore solution to the external surface of the geopolymer concrete [51]. Thereafter, the ions react with atmospheric CO_2_ and water to form a composition of white carbonate salt known as efflorescence [9]. Figure 5 shows the appearance of efflorescence that was only observed on the surface of the GC1–AP0.9–1SS1SH geopolymer concrete mix.

Moreover, the mechanism of efflorescence formation in geopolymer concrete systems differs from that of OPC-based systems. In OPC-based systems, the efflorescence occurs due to the migration of soluble Ca^2+^ ions to the surface and the reaction of such migrated ions with the atmosphere [9]. In contrast to geopolymer systems, efflorescence forms from the migration of Na^+^ and OH^−^ ions from the pore solution to the outer surface and the final reaction with atmospheric CO_2_ and water [7,9]. In practise, the appearance of efflorescence is not aesthetically pleasing and reduces the longevity and strength of building materials [52].

Generally, the effect of the SS/SH ratio observed in geopolymer mix GC0, GC3, and GC4 mixes showed that the UCS value of the geopolymer concretes increased gradually as the SS solution increased at all curing ages. As evidenced in Figure 4, the variation in SS/SH ratio indicated that the geopolymer mixes GC3–AP0.7–1.2SS0.8SH displayed the maximum strength at each curing age. However, when compared with each other, geopolymer mixes GC0–AP0.7–1SS1SH and GC4–AP0.7–0.8SS1.2SH displayed a fluctuating strength development at 7, 14, and 28 day curing ages. An obvious reason could be due to the closeness of the SS solution ratio (1SS and 0.8SS), which does not display a significant difference in UCS. This UCS increase trend indicates the importance of increasing the amount of sodium silicate solution on compressive strength [48]. According to Nath and Sarker [6], this increase in strength is believed to be due to the increase of soluble silica in the system, which modifies the reaction kinetics and ratio of crystallisation by accelerating the polymerisation process and enhancing the condensation process of the dissolved precursor.

#### 4.2.2. Tensile Splitting (TS) Test

TS in concrete is essential for investigating the behaviour of the formulated geopolymer concrete mixes to shear, steel anchorage, crack development, and other applications in hardened concrete [4]. Figure 6 presents the variation in TS strength development for all the concrete mixes at the end of the 28 day ambient curing period. Geopolymer mix GC2 achieved the highest TS strength of 4.2 MPa, while normal OPC (Control 1) and GC1 both experienced the lowest TS strength of 3.3 MPa. It is evident from Figure 6 that the TS strength for each ordinary (OPC-Control 1) and geopolymer concrete mix (GC0–GC4) follows a similar pattern to that of the previously investigated UCS (Figure 2), which shows that the TS strength development is influenced by the A/P and SS/SH ratios. 

## 5. Conclusions

The outcomes from the current study suggest the viability of developing a geopolymer concrete using a laboratory-synthesised silica fume-derived sodium silicate solution in combination with NaOH as an alternative binary alkali-alkaline activator to GGBS, thereby leading to sustainability improvements in construction. Therefore, the following conclusions can be drawn:

The impact of A/P ratio variations in the geopolymer mix GC0, GC1, and GC2 in terms of consistency indicated a reduction in slump values as the A/P ratio decreased from 0.5 to 0.9, while the compatibility degree showed a reversal of the identified trend in slump. Moreover, the variation in the SS/SH ratio (SS increase) at a fixed A/P ratio of 0.7 negatively impacted the consistency of the developed geopolymer concrete (slump reduction) due to the high viscosity of the SS solution;The formulated geopolymer concrete mixes (GC0–GC4) achieved higher UCS at all curing ages than the OPC (Control 1) mix, with mix GC2 attaining the highest UCS value of 62.6 MPa, thereby displaying the superiority of the alkali activation and polymerisation process over the CSH gel formation in ordinary concrete production;The effect of the A/P ratio observed in GC0, GC1, and GC2 mixes in terms of strength showed a UCS value increase at all curing ages (3, 7, 14, and 28 days) for every reduction in the A/P ratio. Therefore, in contrast to the decrease in UCS trend observed with an increase in the A/P ratio, efflorescence, which impacts the aesthetics of the concrete application, is likely to develop on the geopolymer concrete due to the excess alkaline compound in the concrete system;In terms of SS/SH variations, the UCS gain is minimal compared with the A/P ratio variations. This suggests that the A/P ratio has a more pronounced impact on strength gain, which is attributed to the geopolymerisation process;Depending on the concrete design strength or construction applications, a 0.5–0.7 A/P ratio with a SS/SH ratio of 1:1 or a slight SS increase of not more than 2% is advised.Although a laboratory-synthesised SF-derived geopolymer product was successful in this research, certain criteria should be considered in large-scale production on a construction site. As such, for properly storing activators and precursor materials under the required conditions, only machine mixing or ready-mix concrete should be employed because it replicates the mixing regime in the laboratory, and bath mixing operators must have adequate knowledge of geopolymer concrete.

## Figures and Tables

**Figure 1 materials-16-02400-f001:**
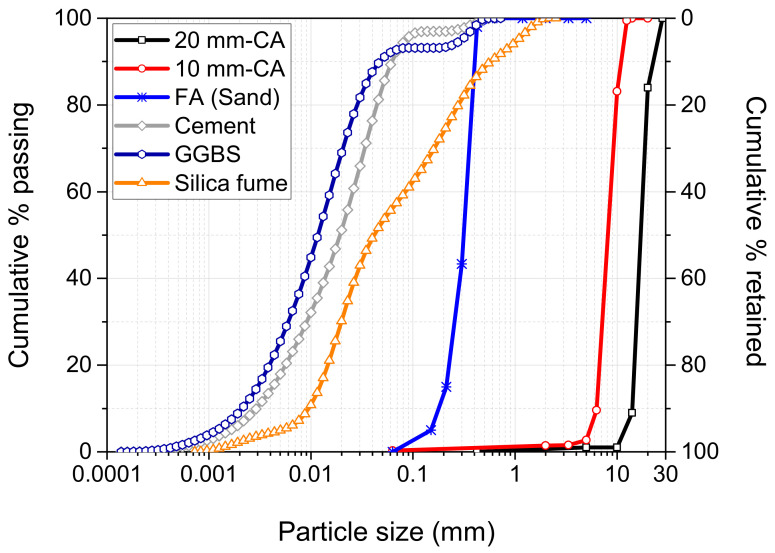
Particle size distribution of the raw materials.

**Figure 2 materials-16-02400-f002:**
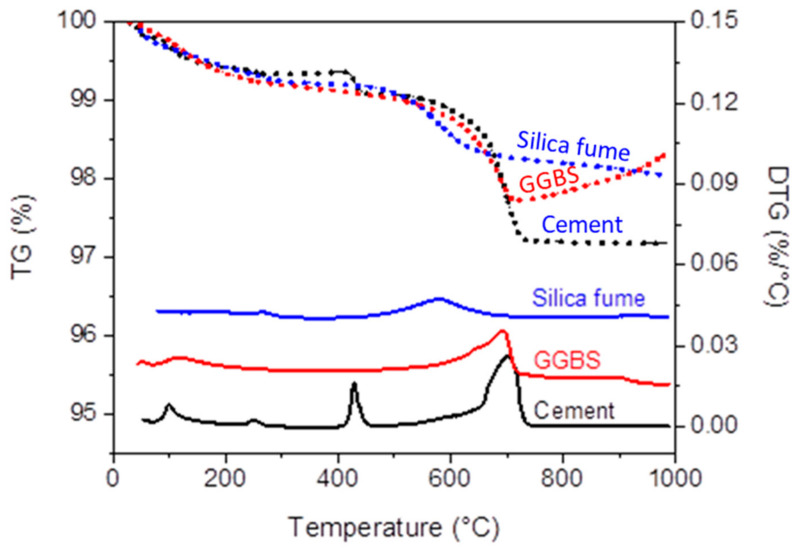
TG (dotted lines) and DTG (solid lines) curves of cement, GGBS, and silica fume.

**Figure 3 materials-16-02400-f003:**
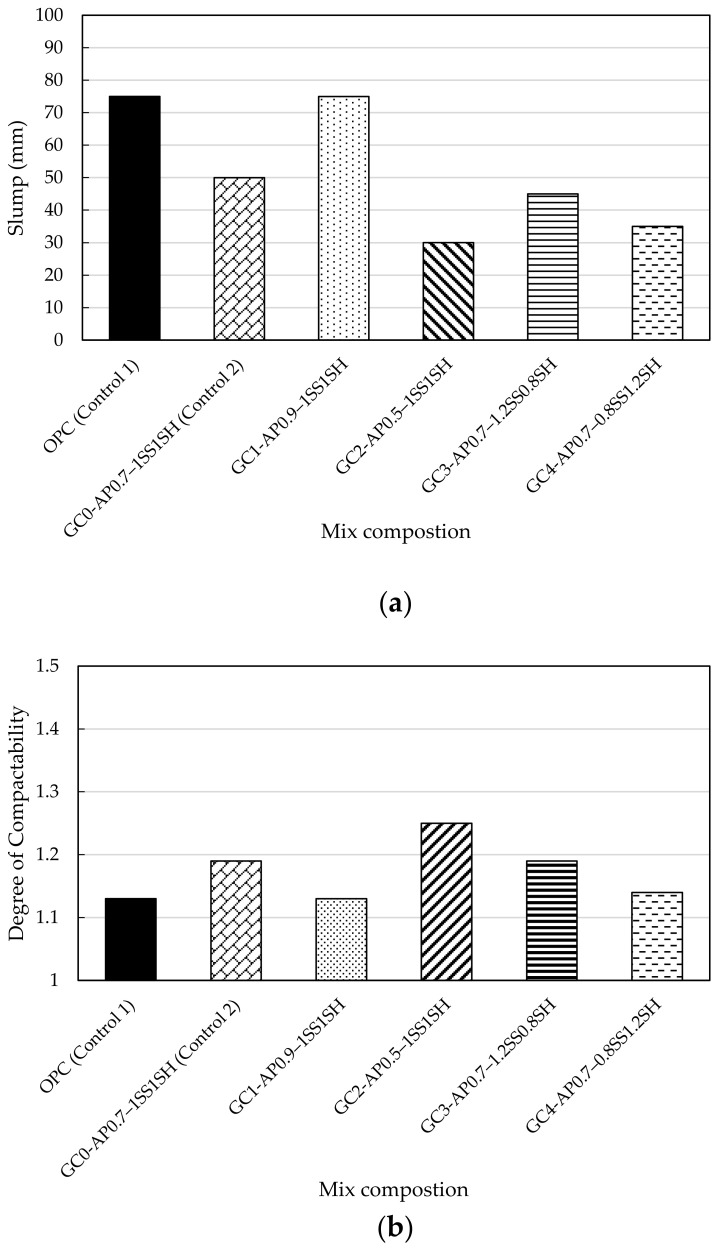
Consistency of concrete mixes measured by (**a**) slump and (**b**) the degree of compactability test.

**Figure 4 materials-16-02400-f004:**
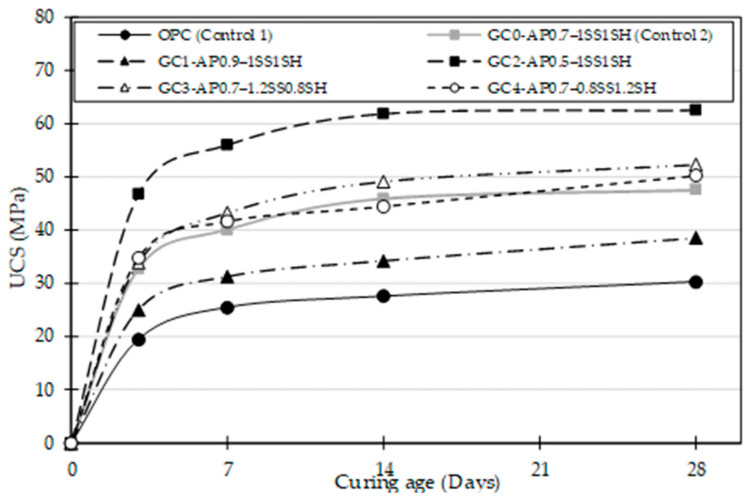
UCS development of the ordinary and geopolymer concrete mixes made with different AP and SS/SH ratios.

**Figure 5 materials-16-02400-f005:**
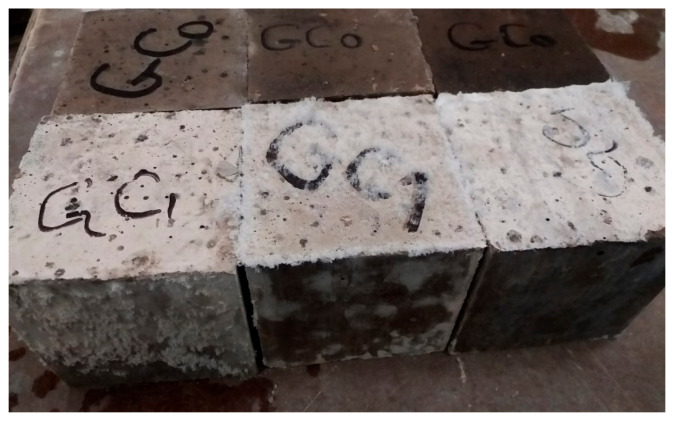
Appearance of efflorescence on the surface of geopolymer concrete specimens with higher alkaline content.

**Figure 6 materials-16-02400-f006:**
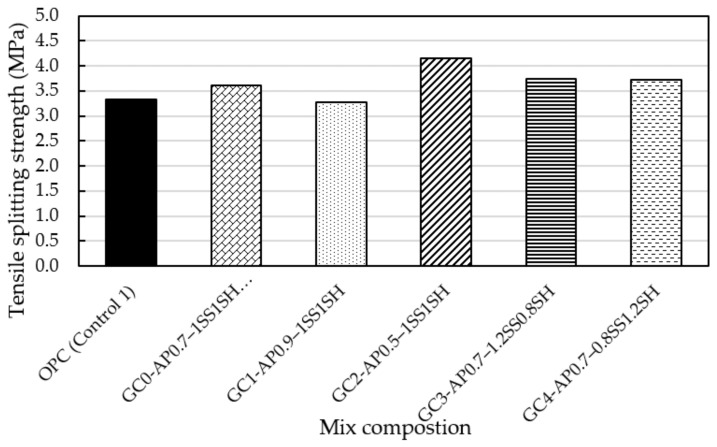
Twenty-eight day tensile splitting strength development of the ordinary and geopolymer concrete mixes made with different AP and SS/SH ratios.

**Table 1 materials-16-02400-t001:** Oxide compositions of OPC, GGBS, and SF.

Oxides	Compositions (%)		
	Cement	GGBS	SF
CaO	61.49	37.99	0.2
MgO	3.54	8.78	0.1
SiO_2_	18.84	35.54	97.1
Al_2_O_3_	4.77	11.46	0.2
Na_2_O	0.02	0.37	-
P_2_O_5_	0.1	0.02	0.03
Fe_2_O_3_	2.87	0.42	0.01
Mn_2_O_3_	0.05	0.43	-
K_2_O	0.57	0.43	0.2
TiO_2_	0.26	0.7	-
V_2_O_5_	0.06	0.04	-
BaO	0.05	0.09	-
SO_3_	3.12	1.54	0.1
Loss on ignition	4.3	2	0.5

**Table 2 materials-16-02400-t002:** Physical compositions of OPC, GGBS, and SF.

Other Properties	Cement	GGBS	SF
Insoluble residue	0.5	0.3	-
Bulk density (kg/m^3^)	1400	1200	300
Specific gravity (Mg/m^3^)	3.15	2.9	3.15
Glass content	-	90	-
Blaine fineness (m^2^/kg)	365	450	-
Alkalinity value (pH)	13.41	10.4	7
Colour	Grey	Off-white	Grey
Physical form	Fine powder	Fine powder	Powder

**Table 3 materials-16-02400-t003:** Some physical properties of the coarse and fine aggregates.

Physical Properties	Coarse Aggregates		Fine Aggregates (Sand)
	20 mm	10 mm	
Uniformity coefficient (C_U_)	1.3	3.3	0.11
Curvature coefficient (C_C_)	7.5	1.5	1.75
Flakiness index (%)	23	30–35	-
Elongation index (%)	12	17–22	-
Shape index (%)	7	12	-
Impact value	15	23	-
Fineness modulus (mm)	-	4	1.54
Uncompacted bulk density (Mg/m^3^)	2.57	1.35	1.5
Pre-dried particle density (Mg/m^3^)	-	2.69	2.6
Water absorption (%)	1.1	2	21

**Table 4 materials-16-02400-t004:** Mix compositions of ordinary concrete (C0) and geopolymer concretes (GC).

Mix Code	Elaborated Abbreviation	Concrete Binder						W (L)	Aggregates (kg)		
		PC (kg)	Geopolymer Binder						FA	10 mm	20 mm
			GGBS (kg)	A/P Ratio	SS:SH	Activator (mL)					
						SS	SH				
C	OPC (Control 1)	7.3	-	-	-	-	-	4	14.6	7.3	14.6
GC0	GC0–AP0.7–1SS1SH (Control 2)	-	4.3	0.7	1:1	1031	1031	1.5	14.6	7.3	14.6
GC1	GC1–AP0.9–1SS1SH	-	3.85	0.9	1:1	1186	1186	1.5	14.6	7.3	14.6
GC2	GC2–AP0.5–1SS1SH	-	4.88	0.5	1:1	835	835	1.5	14.6	7.3	14.6
GC3	GC3–AP0.7–1.2SS0.8SH	-	4.3	0.7	0.8:1.2	1237	825	1.5	14.6	7.3	14.6
GC4	GC4–AP0.7–0.8SS1.2SH	-	4.3	0.7	1.2:0.8	825	1237	1.5	14.6	7.3	14.6

PC—Ordinary Portland Cement; GGBS—Ground Granulated Blast-furnace Slag; A/P—Activator/Precursor ratio; SS:SH—Sodium Silicate to Sodium Hydroxide ratio; FA—Fine Aggregate; W—Water.

## Data Availability

Not applicable.

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
