# Peer review of "Physico-Mechanical Evaluation of Geopolymer Concrete Activated by Sodium Hydroxide and Silica Fume-Synthesised Sodium Silicate Solution"

_materials, 2023, doi:10.3390/ma16062400_

Round 1

Reviewer 1 Report

This is an interesting work. I have only a few notes that would help improve its quality. 

1- There are a few writing issues such as punctuation, grammar, subscripts, ...etc.

2- some related references should be discussed and cited such as the following ... The impact of using rice husk ash as a replacement material in concrete: An experimental study (2022) Journal of King Saud University-Engineering Sciences

3- there are several mistakes in the figures numbers. I believe Line 232 should be Fig 3 (not 6) and the figure on page 8 should also be fig 3 (not 4) 

4- Figures 1 & 2 fit better in the results section 

5- In Figure 2, the TG curves should also be labeled 

Reviewer 2 Report

The authors need to put forth a more convincing reason for using lab synthesized sodium silicate(SS) solution. As silica fume is also a commercially available product, its usage will not lessen the influence of commercial manufacturers of sodium silicate solution on the production of geopolymer concrete. In order to advocate the use of lab synthesized SS solution, one must highlight their advantages (if there are any) in terms of mechanical and durability characteristics as compared to commercially available SS solution based geopolymer concrete. Also, the challenges associated with the production of geopolymer concrete on a mass scale using lab synthesized sodium silicate solution should be highlighted in the introduction.

Author Response

The authors would like to appreciate the constructive suggestions/recommendations from the reviewer, which have been addressed in the revised version of the proposed article with responses. The changes made are marked up for clarity in the revised draft article. 

"Regarding justifying the use of silica fume, the authors have inserted justification comments in two places, firstly  briefly in the abstract and secondly in a slightly expanded form in the introduction."